ecology/environmental science

nature's benefits to people, ecosystem-based adaptation, nature-based solutions, sustainable land management

**Author for correspondence:**
Alanna J. Rebelo
e-mail: arebelo@sun.ac.za

# Benefits of water-related ecological infrastructure investments to support sustainable land-use: a review of evidence from critically water-stressed catchments in South Africa

Alanna J. Rebelo[1], Petra B. Holden[3], Karen Esler[1,2] and Mark G. New[2,4]

[1]Department of Conservation Ecology and Entomology, and [2]DSI-NRF Centre for Invasion Biology, Stellenbosch University, South Africa
[3]African Climate and Development Initiative, University of Cape Town, South Africa
[4]School of International Development, University of East Anglia, Norwich, UK

AJR, 0000-0002-7544-9895; PBH, 0000-0002-3047-1407;
KE, 0000-0001-6510-727X; MGN, 0000-0001-6082-8879

Investments to promote sustainable land-use within critical river catchment areas are often undertaken to provide benefits to society. Investments generally aim to protect or restore ecological infrastructure—the underlying framework of ecosystems, functions and processes that supply ecosystem services—for multiple benefits to society. However, the empirical evidence base from studies across the world on both mechanisms and outcomes to support these assumptions is limited. We collate evidence on the benefits of ecological infrastructure interventions, in terms of ecosystem services provided to society, from three major South African water-providing catchments using a novel framework. In these catchments, millions of US Dollars' worth of investments have been made into ecological infrastructure since 1996. We ask the question: is there evidence that ecological infrastructure interventions are delivering the proposed benefits? Results show that even in catchments with substantial, long-term financial investment into ecological infrastructure, research has not empirically confirmed the benefits. Better baseline data collection is required, and monitoring during and after ecological infrastructure interventions, to quantify benefits to

society. This evidence is needed to leverage investment into ecological infrastructure interventions at scale. Investment at scale is needed to transition to more sustainable land-use to unlock greater benefits to nature and people.

# 1. Introduction

Over the past 50 years, it is estimated that humans have affected 83% of the global terrestrial land surface, with 60% of ecosystem services reported to have declined [1]. There has never been a more pertinent time to find sustainable land-use solutions. Sustainable land-use has been defined as the 'rational development, use and protection of land resources based on specific space–time conditions and adopting appropriate means and organizational forms' [2]. Underpinning sustainable land-use is the protection and restoration of ecological infrastructure which is 'the underlying framework of natural elements, ecosystems, and functions and processes that are spatially and temporally connected to supply ecosystem services' [3]. Interventions to restore or protect ecological infrastructure are wide ranging and context specific, so here we define ecological infrastructure interventions as artificial or natural actions that aim to enhance chosen ecosystem services in intact to transformed landscapes, informed by an understanding of ecology. Examples of natural interventions include alien plant clearing and revegetation, and examples of artificial interventions include artificial wetlands, permeable pavements and erosion control structures (gabions and weirs).

Ecological infrastructure interventions aim to provide multiple ecosystem services to society [4], as well as socio-economic benefits [5]. For example, a global meta-analysis of 89 restoration projects indicated increases in both ecosystem services and biodiversity in restored compared with degraded systems [4]. Ecological infrastructure interventions can also have direct and indirect livelihood benefits, including improved livelihood security and the development of new value chains [5–8]. Restoring degraded ecological infrastructure is additionally thought to improve ecosystem resilience which may buffer society from the impacts of climate change (e.g. the risks of extreme weather events leading to droughts, floods and fires) as well as protecting built infrastructure investments (e.g. protecting water impoundments from siltation through improved, sustainable land-use practices) [7,9]. However, the benefits of investing in ecological infrastructure interventions have been shown to be highly context specific, given the diversity of ecosystems globally [10]. Therefore, this provides justification for the need for a local-scale review of benefits, rather than broad global-scale reviews, which may mask important findings. This too is important for implementation: implementing interventions based on broad global-scale reviews may not be appropriate for local conditions.

In the last decade, more than 80 countries committed to restore over 30 million hectares of degraded land by 2020 under several UN frameworks [11], requiring significant investment into ecological infrastructure. After a decade of investment, is there evidence that ecological infrastructure interventions are delivering the proposed benefits? Additionally, is the evidence base sufficiently distributed in terms of location to support existing investments that are being made across developing and developed nations? Evidence of financial returns as well as improved understanding of investor preferences is suggested to be critical to leverage future investment into ecological infrastructure [12]. Many developing nations face the challenge of leveraging sufficient investment to implement ecological infrastructure interventions at scale [13]. Reasons for this include a lack of evidence of the proposed social and ecological benefits of ecological infrastructure interventions, or a lack of certainty around these proposed benefits [14]. For example, a global review found that the present understanding of the value of wetland restoration is 'tentative' [15]. Despite these challenges, small-scale ecological infrastructure interventions have been growing in popularity globally as forms of sustainable land-use solutions [16]. The lack of context-specific evidence around the benefits of investments into ecological infrastructure may pose problems for securing sufficient funds to upscale these interventions.

In South Africa, a nation with water security challenges, the government has invested in restoring ecological infrastructure with the focus of augmenting water provision and creating jobs [17]. Here, we focus on three major South African water-providing catchments where there has been the most significant investment into ecological infrastructure nationally [13,18]. We collate documented evidence on the ecosystem service and societal benefits of these ecological infrastructure interventions. We ask the question: is there evidence available that ecological infrastructure interventions are delivering the proposed benefits? To support our assessment, we developed an 'ecological infrastructure intervention – ecosystem services' (EII-ES) framework that can be adapted to other

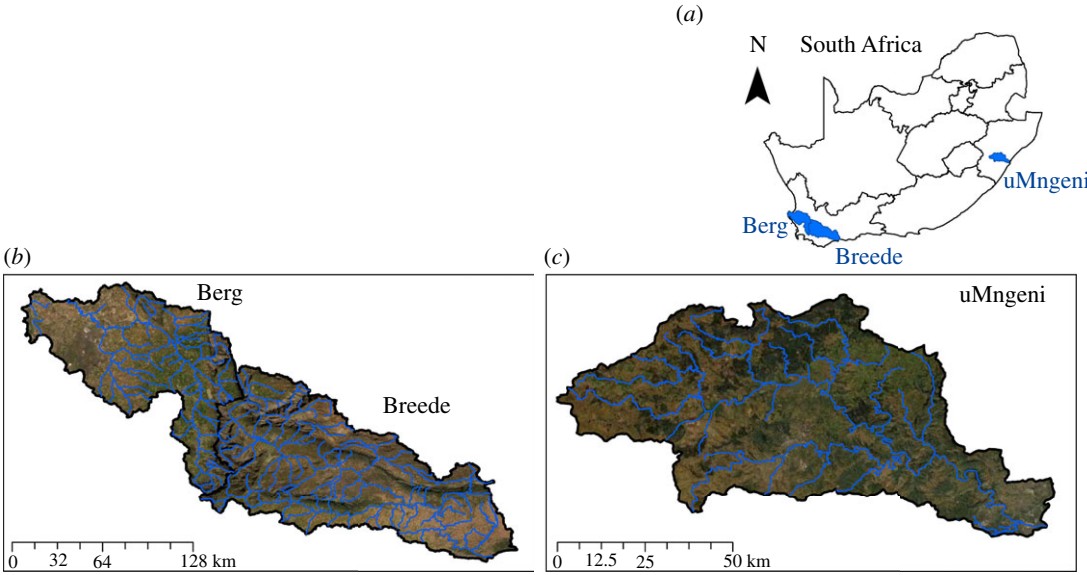

**Figure 1.** (*a–c*) Location of the three study catchments in South Africa, with major rivers overlain on true colour imagery (Berg, Breede and uMngeni).

social-ecological systems. We used this to interrogate the academic and grey literature for evidence of benefits resulting from ecological infrastructure investments. Finally, we describe the integration of the framework with the evidence from the literature review using causal loop diagramming principles.

# 2. Study sites and methodology

## 2.1. Case study catchments

The Berg, Breede and uMngeni catchments are nationally important, water-stressed strategic water source areas [19] (figure 1). Strategic water source areas provide disproportionately high water resources relative to their size, either in terms of surface water runoff or groundwater or both [20]. These three catchments have been selected for this study in particular, due to investments into ecological infrastructure having been made since 1996, to the value of millions of US Dollars. All three catchments are important water sources for large metropoles and important agricultural regions, though the Berg and Breede catchments have quite different socio-ecological characteristics to the uMngeni. Other water-users include agriculture, particularly for the Berg and Breede, as well as industry, predominantly for the uMngeni. The uMngeni catchment is located within the summer rainfall region of South Africa and is composed predominantly of the Grassland Biome. The Berg and Breede catchments are within the winter rainfall region of South Africa and are part of the Fynbos Biome. All catchments are transformed to some degree, and some of the many problems faced include the invasion of alien plants, particularly by alien trees, and also pollution, wetland loss and degradation, and habitat transformation.

## 2.2. Methodology

Our approach to assessing the evidence base in these three catchments included three phases. The first included the development of an 'ecological infrastructure intervention – ecosystem services' (EII-ES) framework which we used to hypothesize the effects of ecological infrastructure interventions on ecosystem processes and services. The second stage included a literature review of peer-reviewed and grey literature in these three catchments. Lastly, we integrated the hypothesized EII-ES framework with the evidence from the literature review using causal loop diagramming principles. We discuss each of these phases in detail below.

### 2.2.1. 'Ecological infrastructure intervention – ecosystem services' framework

We developed a framework within which the effects of different types of ecological infrastructure interventions on ecosystems processes and services were hypothesized (table 1). To do this, the

**Table 1.** The results of applying the 'Ecological infrastructure intervention – ecosystem services' (EII-ES) framework for a generalized South African context. The direction of effects are predicted for various ecological infrastructure interventions on ecosystem services. For the full explanation of the predicted effects, see the electronic supplementary material, table S1. Symbols: ↑ increase, ↓ decrease, ↑↓ both increase and decrease, ↔ no change. Cell colour provides a visual map of these symbols. IAPs: invasive alien plants (with a focus on trees).

| intervention type | chance of success | cost (direct costs) | ecosystem | water flow regulation | | | | regulating services | | | | | provisioning services (fuel, fodder, food, materials) | | cultural services | | | literature review intervention categories (figure 3) |
|---|---|---|---|---|---|---|---|---|---|---|---|---|---|---|---|---|---|---|
| | | | | water purification | water provision | drought protection (sustained baseflow) | flood attenuation | carbon sequestration (soil & veg) | erosion control | biodiversity | pollination (wild & commercial) | employment | commercial | subsistence | tourism ($) | spiritual | aesthetics | |
| agricultural retreat (passive recovery) | low | 0 | mountains hillslopes, riparian zones, buffer zones, wetlands | ↑ | ↑↓ | ↓ | ↔ | ↔ | ↓ | ↑ | ↑ | ↓ | ↓ | ↓ | ↓ | ↓ | ↓ | general 'restoration' |
| reinstatement of natural fire regime | medium–high | 0–low | mountains hillslopes, riparian zones, buffer zones, wetlands | ↑ | ↑↓ | ↑ | ↑ | ↑ | ↑ | ↑ | ↑ | ↑↓ | ↑↓ | ↑↓ | ↑↓ | ↑↓ | ↑↓ | restoring fire regime |
| clearing of IAPs: mechanical, hand pulling | medium | medium | mountains hillslopes, riparian zones, buffer zones, wetlands | ↑ | ↑ | ↑ | ↔ | ↑↓ | ↑↓ | ↑ | ↑↓ | ↑ | ↔ | ↑↓ | ↑↓ | ↑↓ | ↑↓ | clearing of IAPs |
| clearing of IAPs: mechanical, heavy machinery | low | high | mountains hillslopes, riparian zones, buffer zones, wetlands | ↔ | ↑ | ↔ | ↑↓ | ↑↓ | ↑↓ | ↑ | ↑↓ | ↑ | ↔ | ↑↓ | ↑↓ | ↑↓ | ↑↓ | clearing of IAPs |
| clearing of IAPs: biological, fruit/seed attacking | medium | low | mountains hillslopes, riparian zones, buffer zones, wetlands | ↔ | ↔ | ↔ | ↔ | ↔ | ↔ | ↑↓ | ↔ | ↔ | ↔ | ↔ | ↑↓ | ↑↓ | ↑↓ | clearing of IAPs |
| clearing of IAPs: biological, plant attacking | medium–high | low | mountains hillslopes, riparian zones, buffer zones, wetlands | ↑↓ | ↑ | ↔ | ↔ | ↔ | ↑↓ | ↑↓ | ↑↓ | ↔ | ↔ | ↑↓ | ↑↓ | ↑↓ | ↑↓ | clearing of IAPs |
| clearing of IAPs: chemical, direct application (stump & local spray) | low–medium | medium | mountains hillslopes, riparian zones, buffer zones, wetlands | ↑↓ | ↑ | ↑↓ | ↑↓ | ↔ | ↑↓ | ↑ | ↑↓ | ↑ | ↔ | ↑↓ | ↑↓ | ↑↓ | ↑↓ | clearing of IAPs |

**Table 1.** (*Continued.*)

| | | | | | | | | | | | | | | | | |
|---|---|---|---|---|---|---|---|---|---|---|---|---|---|---|---|---|
| clearing of IAPs: chemical, foliar/aerial spraying | low | high | mountains, hillslopes, riparian zones, buffer zones, wetlands | ⇄ | ← | ↕ | ↕ | ↕ | ↕ | ↕ | ↕ | → | → | → | → | clearing of IAPs |
| revegetation with irrigation; seeds | medium | high | mountains, hillslopes, riparian zones, buffer zones, wetlands | ↕ | ↕ | ↕ | ↕ | ↕ | ↕ | ↕ | ← | ← | ← | ⇄ | ⇄ | revegetation |
| revegetation with irrigation; adult plants | medium | high | mountains, hillslopes, riparian zones, buffer zones, wetlands | ← | ↕ | ↕ | ↕ | ↕ | ↕ | ↕ | ← | ← | ⇄ | ⇄ | ⇄ | revegetation |
| revegetation without irrigation; seeds | low | medium | mountains, hillslopes, riparian zones, buffer zones, wetlands | ↕ | ↕ | ↕ | ↕ | ↕ | ↕ | ← | ← | ← | ⇄ | ⇄ | ⇄ | revegetation |
| revegetation without irrigation; adult plants | low | medium | mountains, hillslopes, riparian zones, buffer zones, wetlands | ↕ | ↕ | ↕ | ↕ | ↕ | ↕ | ← | ← | ← | ⇄ | ⇄ | ⇄ | revegetation |
| rehabilitation (gabion/weir) | low–medium | very high | mountains, hillslopes, riparian zones, buffer zones, wetlands | ↕ | ⇄ | ⇄ | ⇄ | ⇄ | ⇄ | ⇄ | ← | ← | ← | ← | ← | rehabilitation (gabion/weir/dam) |
| construction of artificial wetlands | high | high | wetlands | ← | ↕ | ↕ | ↕ | ↕ | ↕ | ↕ | ↕ | ← | ⇄ | ⇄ | ⇄ | artificial wetland |
| removal of solid waste | high | medium | riparian zones, wetlands | ← | ↕ | ↕ | ↕ | ↕ | ↕ | ↕ | ↕ | ← | ← | ← | ← | - |
| combo: agricultural retreat AND revegetation with irrigation; adult plants & seeds | high | high | mountains, hillslopes, riparian zones, buffer zones, wetlands | ← | ⇄ | ← | ← | ← | ← | ← | ⇄ | ⇄ | ⇄ | ⇄ | ⇄ | - |
| combo: clearing of IAPs; mechanical, hand pulling AND revegetation with irrigation; adult plants & seeds | high | high | mountains, hillslopes, riparian zones, buffer zones, wetlands | ← | ← | ← | ← | ⇄ | ⇄ | ← | ⇄ | ⇄ | ⇄ | ⇄ | ⇄ | restoration: clearing of IAPs and revegetation |

direction of the impacts on ecosystem services by specific ecological infrastructure interventions was extracted from different evidence streams: literature, workshops and expert knowledge (table 1). An explanation of the reasons and processes behind the direction of impact was also given (electronic supplementary material, table S1) for the specific context of these South African catchments.

Several assumptions were made to establish the direction of impacts: (i) that the interventions will result in natural ecosystem recovery, the extent of which depends on probable chance of success of the intervention (table 1), (ii) that some time has passed (restoration often has lag effects, and we hypothesize about the overall net effect, not immediate impact), (iii) that a natural fire regime applies (in practice this may not be the case), (iv) in the case of 'revegetation' as an intervention, that either invasive alien trees have been cleared already, or that there were none to begin with, (v) also for revegetation that there is sufficient mulch and soil organic carbon for native seedling establishment, and (vi) in terms of the building of gabions or weirs, we consider impacts both above and below the structures [21].

### 2.2.2. Literature review

There has been substantial financial investment into ecological infrastructure interventions in the Berg, Breede and uMngeni catchments in South Africa in comparison to other catchments [13,18]. Therefore, we aimed to establish whether there is evidence available that ecological infrastructure interventions are delivering the proposed benefits, to support the investments that have been made in these catchments. The literature review was conducted in January 2019 for both peer-reviewed (ScienceDirect) and grey literature (using the South African Water Research Commission Knowledge Hub—a catalogue of reports on water-related research funded by the Water Research Commission, a National government institution—and Google searches). The literature review had both a systematic and non-systematic component. The non-systematic component was driven by documents and literature obtained through engagements with local researchers and practitioners in the field of restoration or protection of ecological infrastructure in the catchments. For the systematic component, search terms included the catchment name (i.e. 'Berg', 'Breede' and 'uMngeni/uMgeni') and 'ecological' and 'infrastructure' for the first search. The terms 'rehabilitation' and 'restoration' with the catchment names were used for the second and third searches, respectively. The full text of the articles was scanned for these search terms.

The total papers/reports for each site were: Berg (1029), Breede (502) and uMngeni (113). These were given an initial screening for relevance (criteria were that some ecological infrastructure intervention was investigated, and some benefit was described). Three hundred and ten papers and reports were selected for further analysis. Approximately 58% of studies were from the three catchments, approximately 11% elsewhere in South Africa, and approximately 20% in the rest of the world (figure 2). The publications from outside the study catchments are from the non-systematic component. Of the total selected, 190 papers/reports related to ecological infrastructure. Of these 190, 61% specifically considered an intervention ($n = 106$) and 56% specifically mentioned a benefit in terms of ecosystem service delivery ($n = 69$). Of these 69 papers/reports that specifically mentioned a benefit, 74% explained the process behind this (i.e. how the intervention affected ecosystem service delivery) ($n = 51$). These 69 papers/ reports were selected for further, detailed analysis.

In the more detailed analysis, the specific ecological infrastructure interventions of these 69 papers/ reports were recorded, as well as the ecosystem process and/or ecosystem services investigated. This resulted in 393 cases of reported benefits of investing in ecological infrastructure. It is important to note that sometimes the linkages were implied and not explicit. For example, because the concept of ecological infrastructure, and even ecosystem services, is relatively new [22,23], many of the older publications did not explicitly indicate that they were investigating 'ecosystem services' or 'ecological infrastructure'. For the ecosystem processes and services, the direction of change was recorded (i.e. increase, decrease, both, no change), as well as the type of evidence (i.e. empirical, modelled, conceptual). For empirical and modelling studies, the confidence of the method was rated from 0 (low) to 1 (high) based on the rigour of the methods used. Three criteria were used to judge rigour: (i) the appropriateness of the indicator used (i.e. for ecosystem services: carbon sequestration would be considered only to quantify one aspect of the climate regulation ecosystem service) [22], (ii) whether the methodology was thorough (i.e. as opposed to a rapid assessment, in other words, was spatial and temporal variation considered?), and (iii) whether uncertainty was quantified or not, and for modelling, whether validation was done.

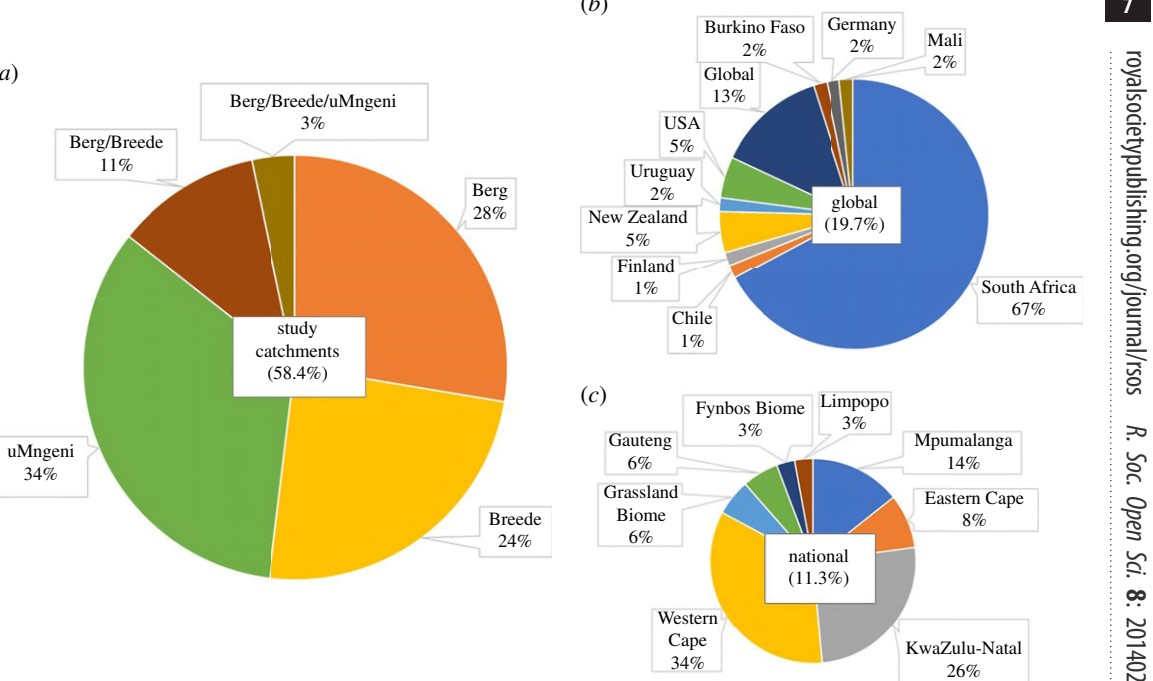

**Figure 2.** The spatial breakdown of literature included in the review ($n = 310$): (*a*) 58.4% of the studies were from the Berg, Breede or uMngeni, (*c*) 11.3% were from one of the South African provinces or biomes, and (*b*) 19.7% were from other countries or at a global scale; 10.7% of studies were conceptual (not shown).

From the 393 cases, lists of ecological infrastructure interventions ($n = 12$), ecosystem properties and processes ($n = 34$) and ecosystem services ($n = 19$) were generated. Next, the 269 cases relating to ecosystem services were extracted for further analysis (there were 179 cases relating to ecosystem properties, as there was some overlap where ecosystem properties had corresponding ecosystem services measured). The terminology for the ecosystem services framing was based on Boerema *et al.* [22]. The 12 interventions, listed in figure 3, are self-explanatory except for the category: 'restoration' which needs further description. Due to multiple studies referring to 'restoration' in either a vague or holistic sense, we included this as a general category. There are two other 'restoration' categories which are clearly described by their names. The general 'restoration' category also included any combination of interventions, including for example: maintenance of ecological infrastructure, sustainable agricultural and grazing management, soil and water conservation activities, invasive alien plant clearing, restoring hydrological functioning, rehabilitating wetlands, and implementing prescribed burning. As a result of these complex combinations of interventions (more 'holistic' approaches), we could not assign these cases to any of the other ecological infrastructure interventions categories.

Rose charts were used to illustrate the $\log_{n+1}$ of the number of cases, the benefit of implementing each ecological infrastructure intervention and the type of evidence supporting this [24,25]. For both empirical and modelling studies, means of confidence scores were calculated and displayed on the plots. The ecosystem process or property measured by each study, as well as the list of citations recorded per ecosystem service are available in the electronic supplementary material, table S2.

### 2.2.3. Systems dynamic conceptual model

The EII-ES framework developed was used to build a systems dynamic conceptual model of ecosystem properties (stocks), processes and ecosystem services using systems thinking and causal loop diagramming principles [26]. This was an initial model based on our hypothesized understanding of how ecosystem properties (stocks), processes and ecosystem services interact in relation to investments in ecological infrastructure interventions. This model was then refined using the results from the review of the 69 studies (393 cases), described above. For all links involving empirical evidence, we represented the arrows in bold. We use the model to highlight current understanding as well as

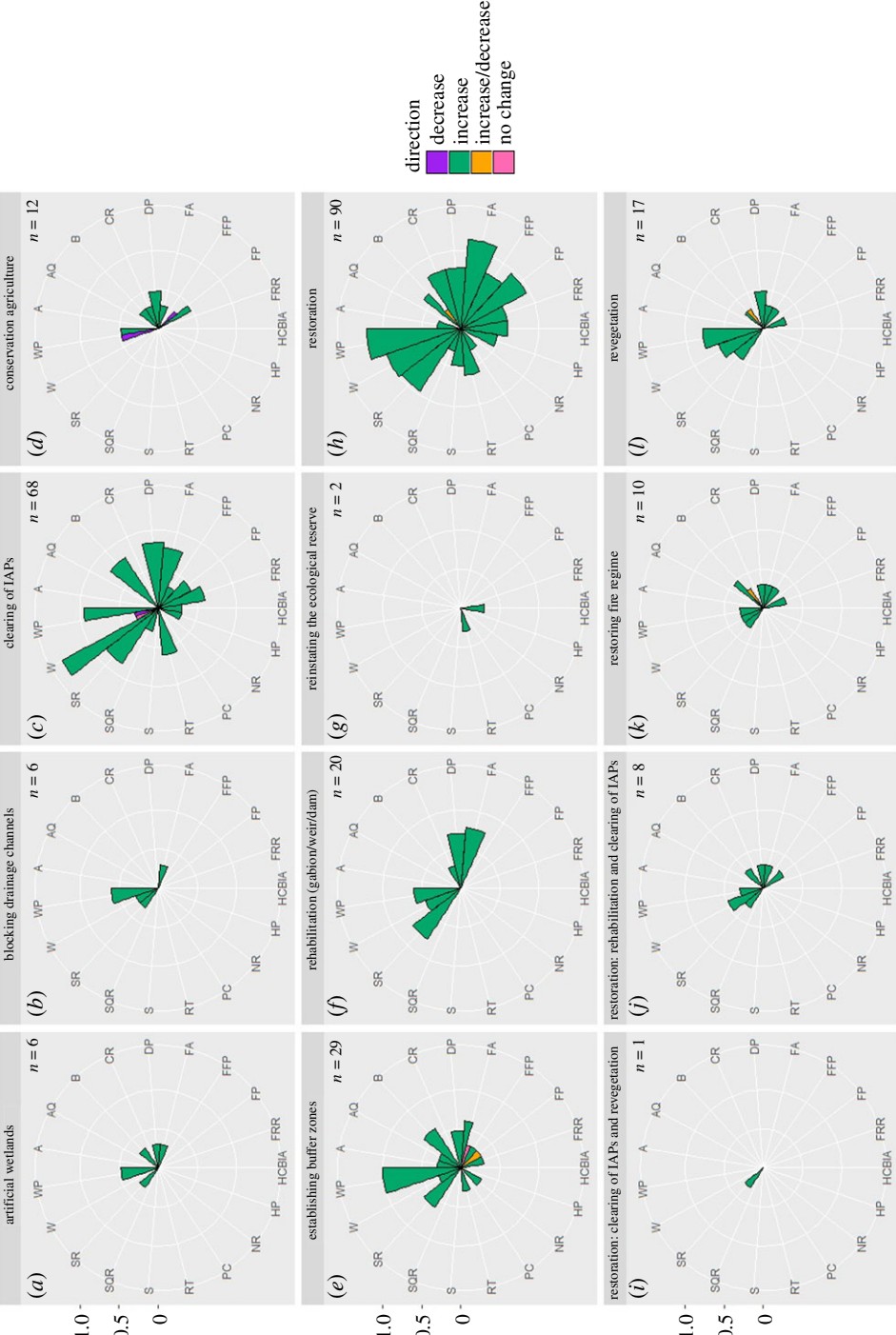

**Figure 3.** (*a*–*l*) Rose charts of the $\log_{n+1}$ of the number of cases indicating the direction of the benefit of implementing each ecological infrastructure intervention according to the literature for the study catchments. The number of cases for each intervention is indicated on each plot using the following symbols: A = Aesthetics, AQ = Air Quality Regulation, B = Biodiversity, CR = Climate Regulation, DP = Drought Protection, FRR = Fire Risk Reduction, FA = Flood Attenuation, FP = Food Production, FFP = Fuel and Fibre Production, HP = Habitat Provision, HCBIA = Heritage, Cultural, Bequest, Inspiration and Art, NR = Noise Reduction, RT = Recreation and Tourism, SR = Sediment Retention, SQ = Soil Quality Regulation, S = Spiritual, W = Water Provision, WP = Water Purification and PC = Pest Control.

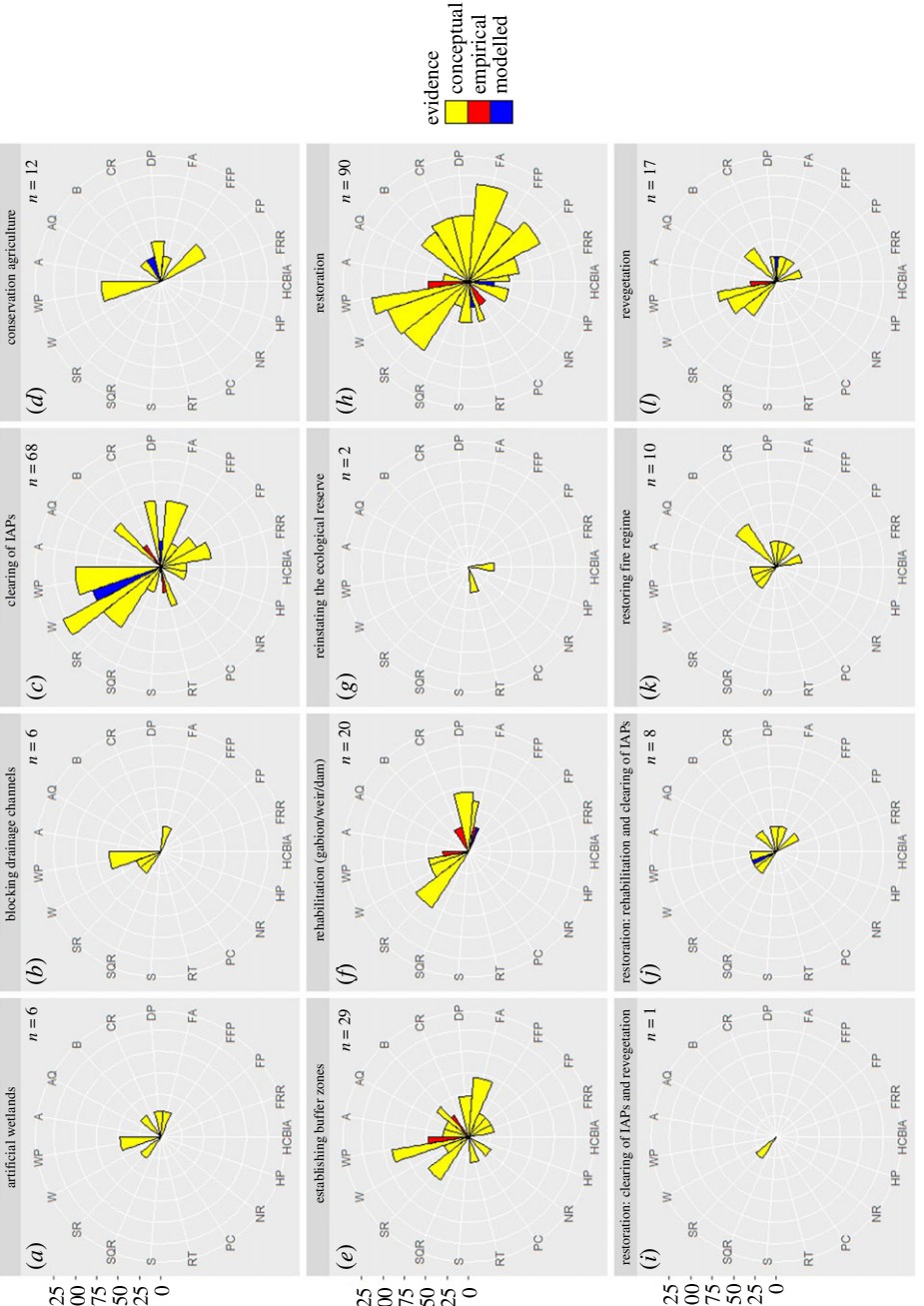

**Figure 4.** (*a–l*) Rose charts of the $\log_{n+1}$ of the number of cases per type of evidence given for the benefits of implementing each ecological infrastructure intervention in the study catchments according to the literature. The number of cases for each intervention is indicated on each plot (*n*), and for empirical and modelled evidence, an average certainty is given on the plot, ranging from 0 to 1. Ecosystem services are indicated on the plot using the following symbols: A = Aesthetics, AQ = Air Quality Regulation, B = Biodiversity, CR = Climate Regulation, DP = Drought Protection, FA = Flood Attenuation, FP = Food Production, FFP = Fuel and Fibre Production, FRR = Fire Risk Reduction, HP = Habitat Provision, HCBIA = Heritage, Cultural, Bequest, Inspiration and Art, NR = Noise Reduction, RT = Recreation and Tourism, SR = Sediment Retention, SQ = Soil Quality Regulation, S = Spiritual, W = Water Provision, WP = Water Purification and PC = Pest Control. IAPs = Invasive alien plants.

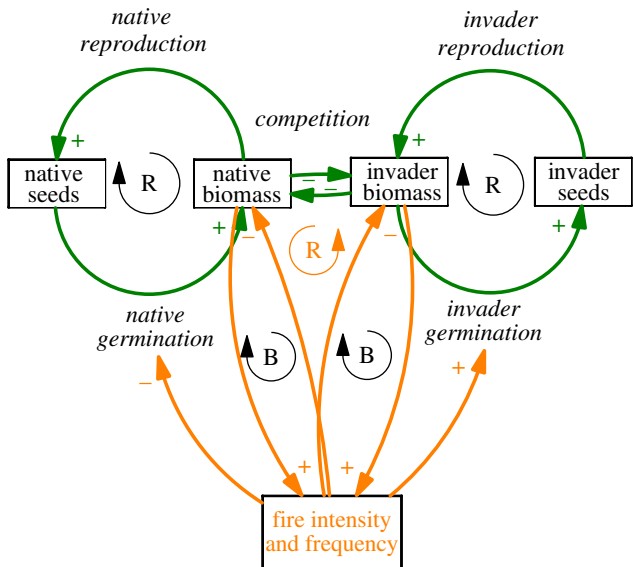

**Figure 5.** A causal loop diagram showing balancing and reinforcing loops associated with natural fire-adapted ecosystems (on the left) and alien invaded ecosystems (on the right), specifically fire-adapted trees. The thickness of the lines indicates that this conceptual alien invasion is aggressive and successful, favouring a trajectory towards perpetuated invasion indicated by the reinforcing loop given in orange (increased invader biomass leads to increased fire intensity and frequency, which negatively impacts native seed germination, resulting in lower native biomass, which creates less competition for invaders). Fire dynamics are shown in orange. Stocks are in boxes, processes are in italics (adapted from Gaertner *et al.* [30]).

knowledge gaps in the linkages between ecological infrastructure interventions, ecosystem properties (stocks), processes and ecosystem services for the three study catchments.

# 3. Results

## 3.1. 'Ecological infrastructure intervention – ecosystem services' framework

For the EII-ES framework, we identified 17 different types of ecological infrastructure interventions and assessed the chance of success and relative cost of each intervention for the study catchments (table 1). There is often a positive relationship between the direct costs and chance of success, but not always; a notable example includes the reinstatement of natural fire regimes. In South Africa, many landowners manipulate the fire regime for rangeland benefits, including increasing the return interval to favour pioneer grasses for grazing [27,28]. To return the fire interval to a natural average would have no direct costs and may even incur direct savings, however, there may be indirect costs in terms of agricultural productivity.

We hypothesized the impacts of these 17 interventions on 14 different ecosystem services for the study catchments (table 1). The reasons and processes behind the direction of impact are given in the electronic supplementary material, table S1 for the specific context of these South African catchments. The 17 interventions and 14 ecosystem services resulted in 238 interactions. The majority of ecosystem services were hypothesized to increase as a result of interventions (30%) or result in an increase and/ or no change (10%), followed by both increases and decreases together (25%), no change (24%) and decreases (6%) or decrease together with no change (5%).

## 3.2. Literature review

From the literature review, we identified 12 broad groups of ecological infrastructure interventions, which partly mapped out onto the 17 types from the EII-ES framework (table 1 and figure 3), but with some clear differences. The interventions identified from the literature tended to be substantially less specific. Overall, seven interventions were in common. The interventions from the EII-ES framework separated alien clearing into more detailed interventions (e.g. by the different methods,

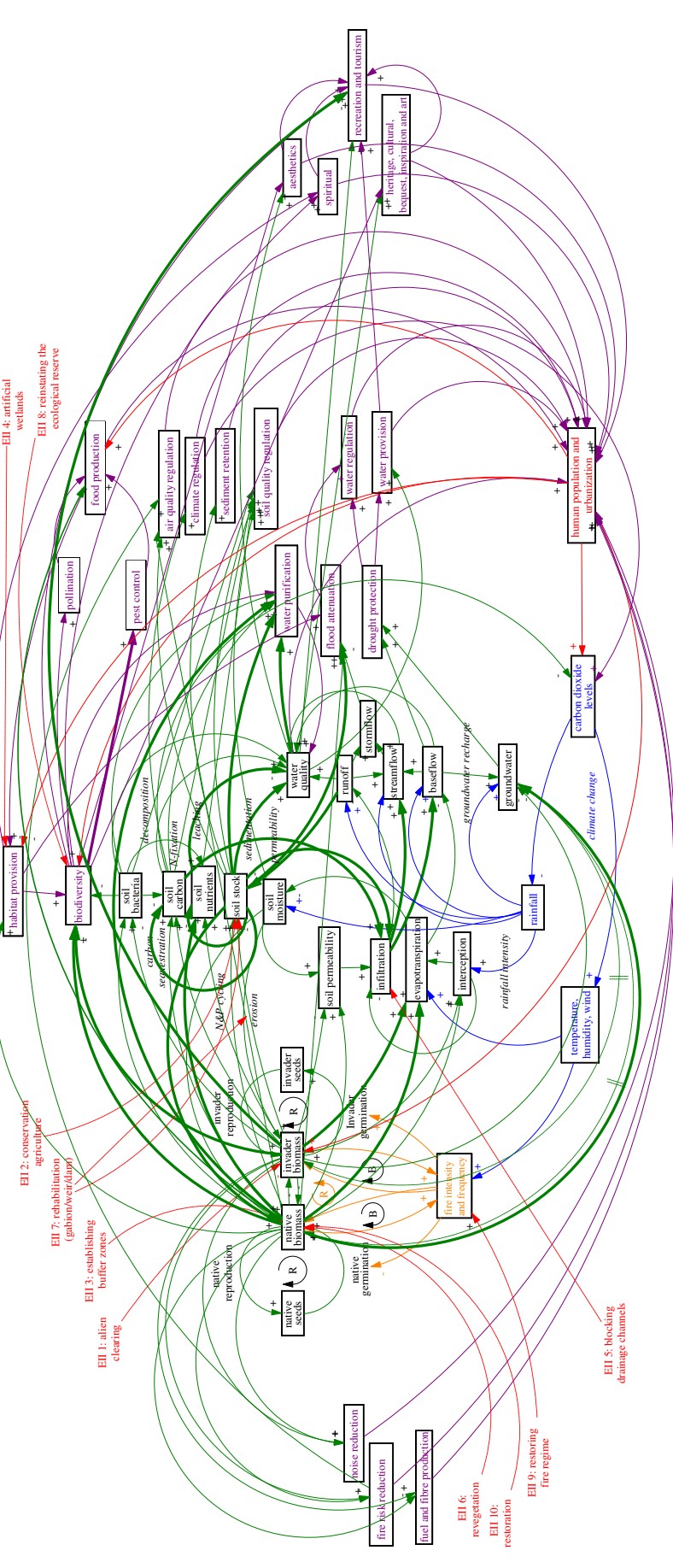

**Figure 6.** Conceptual framework of the impacts of ecological infrastructure interventions. The potential (not-exhaustive) relationships between ecosystem properties, processes and ecosystem services are shown for the South African case studies. The causal loop diagram shows linkages between native-invader dynamics (figure 5) and internal ecosystem properties (stocks in black boxes with green arrows), processes (black italics), ecosystem services (purple) and external anthropogenic factors (red). The climate system is shown in blue, and fire dynamics in orange (some elements adapted from Luvuno et al. [31]). Ecological infrastructure interventions (EII) are given in red, labelled EII1–10 (no boxes), demonstrating the impacts that investments in ecological infrastructure could have on various key ecosystem processes and services. Links (arrows) where empirical evidence exists to support the relationship between variables for these South African case studies are given in bold.

including biological control), whereas the literature tended to be more general, referring to alien clearing broadly, with no mention of biological control, resulting in one broad category. Another major difference is that the EII-ES framework did not include conservation agriculture, but instead agricultural retreat, the latter more from payments for ecosystem services framing (cessation of traditional agriculture), the former considering the derivation of some agricultural benefits combined with more sustainable land-use.

The 12 types of ecological infrastructure interventions identified from the literature can be divided into four main categories in terms of their targeted impacts: the first are those that aim to restore native plant biomass (establishing buffer zones, revegetation, 'restoration', restoring the fire regime), the second are those that target reducing alien plant biomass (alien clearing, and the general 'restoration' category), the third are those that target sediment retention and influence system hydrology (rehabilitation, conservation agriculture, blocking drainage channels) and the fourth are those that aim to impact fauna, by providing habitat, or restoring abiotic cues (e.g. artificial wetlands and the reinstating of the ecological reserve [29]).

Of the 12 broad types of ecological infrastructure interventions, invasive alien plant clearing, establishing buffer zones and the general 'restoration' category had the largest evidence base, with more than 25 cases investigating their benefits (figure 3). In most cases (96%), the direction of change to ecosystems services as a result of implementing an ecological infrastructure intervention is reported as positive (increasing). There are only a few cases of reported decreases (1.5%), both increase and decrease (1.5%), and no change (1%). However, for all interventions, most of the evidence is propositional (i.e. stated or described, but not measured). For example, of the total of 269 cases (69 papers/reports), only 12 involved direct (empirical) measurements (4%), 13 were modelled (5%) and 244 were conceptual (91%) (figure 4). In terms of confidence, approximately 71% of empirical and modelling studies scored high confidence (i.e. 0.8–1), approximately 24% scored low (i.e. 0–0.3) and approximately 6% fell in between (0.4–0.7).

## 3.3. Integration of 'Ecological infrastructure intervention – ecosystem services' framework and literature review

Using the systems dynamic model, we show that five of these ecological infrastructure interventions (and two of the four categories described above) are implemented at the interface of a native versus alien biomass dynamic feedback loop (figure 5). Woody alien invasion in both the Berg, Breede and uMngeni catchments is aggressive and successful, favouring a trajectory towards perpetuated invasion indicated by the reinforcing loop. Increased invader biomass leads to increased fire intensity and frequency, which negatively impacts native seed germination, resulting in lower native biomass, which creates less competition for invaders (Gaertner *et al.* [30]). This native versus alien biomass dynamic feedback loop has cascading impacts on the entire ecosystem, affecting soils, hydrology and biodiversity and ecosystem services (figure 6). The systems dynamic model shows that the links between ecosystem properties (stocks) and processes tend to be relatively well established for these catchments (arrows in bold); however, the relationships between these ecosystem properties and ecosystem services (i.e. benefits to people) are poorly established. The weight of evidence is biased towards understanding the impacts of interventions on ecosystem properties and ecosystem services, rather explicitly making the link to implications for society (figure 6).

## 3.4. Summary of main findings

The key results from the three analyses are summarized as follows. Our novel EII-ES framework hypothesized that 40% of ecosystem services were likely to either increase, or increase together with no change, 25% were hypothesized to either increase or decrease depending on the context, while 24% and 11% were hypothesized to not change, or decrease together with no change, respectively. This differed for the literature review, where the vast majority of cases (96%) showed increases in ecosystem service provision. This suggests a bias, whereby only the ecosystem services thought to increase are investigated or considered in research. Only 4% of studies reviewed involved direct measurements, while 5% were modelled, and the vast majority were conceptual (91%). The quality of the evidence was relatively high for 71% of empirical and modelling studies. There was little overlap between the interventions considered in the EII-ES framework and those that emerged from the literature (seven types were in common). The systems dynamic model highlighted some important

knowledge gaps, revealing that the benefits of ecological infrastructure investments were only empirically measured on five ecosystem services as well as biodiversity, whereas the impacts on ecosystem properties are considerably better quantified.

# 4. Discussion

Overall, the results suggest that the state of the ecological infrastructure interventions evidence base is empirically limited, and this has important implications. In terms of sustainable land-use, it appears that in these three South African catchments, ecological infrastructure interventions are mostly based on sound ecological principles; however, the empirical evidence on process and outcomes is lacking. This makes it difficult to sustain good management practices and investments into ecological infrastructure. We discuss the results and their implications further here, in three subsections: §4.1. The state of the ecological infrastructure interventions evidence base, §4.2. The importance of local, empirical studies and §4.3. The evidence base to leverage investment.

## 4.1. The state of the ecological infrastructure interventions evidence base

We aimed to establish whether there is evidence available that ecological infrastructure interventions are delivering the proposed benefits in three well-invested and researched South African catchments. The main finding of this review is that the evidence for the benefits of ecological infrastructure interventions in these catchments is mostly propositional. Although most of the research shows positive benefits, 91% of this research to date has been conceptual. This result is similar to that of an international review on economic evaluation of wetland restoration that concluded that the understanding of the value achieved is 'tentative' [15]. The state of the evidence base of ecological infrastructure investments in these three catchments is that most evidence is propositional, with very few empirical studies (4%, with an additional 5% modelling studies), and those that exist do not sufficiently make the link to benefits for people. Many of the more conceptual studies also tend to refer to vague or broadly defined interventions (i.e. 'restoration' in general).

In the more-established field of ecological restoration (included under the definition of 'ecological infrastructure interventions'), where clear methods and indices exist for monitoring and evaluation, a global review found that the number of empirical evaluations has grown substantially, as have those that assess ecological functions [32]. However, we identified gaps when it comes to measures of socio-economic attributes and quantifying ecosystem services. For instance, empirical studies (as opposed to conceptual and modelling studies) only quantified the impacts of interventions on five ecosystem services: flood attenuation, recreation and tourism, water purification, pest control, and a component of (or process underpinning) climate regulation (i.e. carbon sequestration). Impacts on biodiversity, which has been shown to underpin ecosystem service provision, were quantified by two studies [22,33]. Therefore, the impacts of these interventions on many ecosystem services remain unexplored, as do the implications (i.e. economic or social benefits) for society.

Although the ecological infrastructure investment evidence-based is limited for these three catchments, in most cases ecological infrastructure interventions were found to be beneficial. A review of nature-based solutions in the United Kingdom also found that, while there is a clear evidence base linking certain interventions to human well-being, there is no comprehensive mapping of multiple positive and negative outcomes for different contexts [34]. Therefore, anecdotal and scattered evidence of benefits exists, but clear empirical evidence is needed to leverage previously untapped investments [12]. To leverage sufficient investment for the implementation of ecological infrastructure interventions at scale [35], we recommend more focused and applied research into the benefits of ecological infrastructure interventions. 'Focused' suggests that the intervention type, scale, scope and ecological context of the study should be clearly defined, and 'applied' research suggests that the link to benefits to people should be explicitly made.

## 4.2. The importance of local, empirical studies

The complexity of ecosystems and the social contexts in which they are embedded implies that benefits of ecological infrastructure investments are likely to be highly context-specific [10]. Therefore, local empirical studies, rather than broad conceptual studies or reviews, are critical to accurately test and measure the projected impacts of ecological infrastructure interventions. Only then can inappropriate

generalizations and extrapolations of the impacts of interventions on ecosystem processes and their services be avoided. One critical and current example is the potential impact of large-scale afforestation programmes, climate adaptation policies that are inappropriately applied to grasslands, savannahs and shrublands, and projected to destroy valuable ecosystems [36]. These poorly designed climate adaptation policies may be because local specialists were not involved in the process, and/or they are applying concepts or interventions developed in specific contexts to inappropriate ecosystems [37]. A potential concern for the relatively young field of ecological infrastructure is that there has been a noticeable shift globally in favour of modelling and data analysis studies, where fieldwork studies are declining [38,39]. In the field of conservation science, a bias was found among publications, with top-ranking academic journals publishing fieldwork studies less frequently than lower ranking journals [38]. This is thought to be due to both bottom-up (e.g. publishing and academic reward systems) and top-down pressures (e.g. current societal needs/priorities) [38]. There have been calls to shift this trend, as fieldwork is acknowledged to form the foundation for model-building in ecology [39], as well as to underpin policy and practice [38].

The systems dynamic model constructed in this study for a 'generalized South African ecosystem' demonstrates complexity through the interconnectedness of each property and process in the ecosystem. It highlights how an investment that targets a single ecological infrastructure intervention in isolation (e.g. alien clearing) is unlikely to be successful in a complex, fire-adapted ecosystem. The complexities of fire-adapted ecosystems (e.g. in terms of seedbank dynamics, native species re-establishment and biotic and abiotic thresholds) means that holistic approaches (e.g. also considering ecological or restoration burns and active restoration) are critically important for successful ecological infrastructure interventions [40,41]. One ecological infrastructure intervention (e.g. clearing alien trees) applied without considering a more holistic approach (e.g. considering fire dynamics or the potential need for an active restoration) has often exacerbated problems [40]. For instance, it has been shown that within a single intervention (i.e. alien clearing), spatial complexities need to be considered as the invasion will play out at different stages across a landscape [41]. The complexity of these ecosystems also suggests that the likelihood of intervention success will probably decrease with time since infestation, or at least the cost of active interventions will increase, due to the impact of the invasive biomass on soils (abiotic thresholds) and depletion of soil seedbanks of native species (biotic thresholds). These insights indicate a need for locally appropriate, holistic, strategic and adaptive approaches to research on and monitoring and evaluation of ecological infrastructure interventions. The systems dynamic model also highlights the gaps in our knowledge for these three catchments (non-bold arrows), specifically how certain ecological infrastructure interventions may have a ripple-like impact through the ecosystem to society. In addition to focused and applied research, we recommend local, empirical research (i.e. fieldwork) to help build an evidence base for the benefits of ecological infrastructure investments.

## 4.3. The evidence base to leverage investment

To leverage investment into ecological infrastructure for more sustainable land-use, a better understanding of financial returns as well as investor preferences has been cited [12]. In South Africa, most research into ecological infrastructure investment currently only considers economic welfare benefits and, therefore, is targeted towards the public sector and grant funding streams [42–44]. There has been little research into explicit benefits generating financial returns from ecological infrastructure interventions, which could leverage private capital. However, in the case of the three catchments, there has already been significant government investment into ecological infrastructure over the past decade [13,18], but little private sector investment. It would therefore appear that although government and private sector investment require similar or overlapping types of evidence, the quality of evidence needed for private sector investment (and therefore to estimate the financial returns) may be higher. However, empirical field-based research does not seem to have been prioritized for these catchments over the last decade, despite significant public sector investment.

Could the lack of empirical evidence at a local scale be hindering potential large-scale private sector investment that is currently missing in South Africa, and globally? Despite significant public sector funding, the overall value and scale of investments into ecological infrastructure in South Africa is considered small relative to that required [35,45]. Investment at scale is needed to promote holistic, ecologically appropriate, strategic and adaptive interventions, to prevent piecemeal, damaging approaches, and to generate measurable impacts. Despite over-burdened public sector budgets and a funding gap, there has been limited effort to quantify benefits, especially in terms of ecosystem

service supply to society, and this is the case for the three catchments in South Africa with the greatest investments into ecological infrastructure to date. This lack of evidence after a decade of investment poses a barrier for leveraging private sector investment [12]. Internationally, there have been several innovative mechanisms proposed to mobilize private sector capital at scale, primarily through demonstrating a financial return on investment [46,47]. We recommend that government makes a portion of the funding set aside for implementation available to support focused, applied and local empirical research to build an evidence base for the benefits of ecological infrastructure investments, thereby potentially relieving constrained public sector budgets in the future.

# 5. Conclusion

The results of this study support five main conclusions: (i) The evidence base for the benefits of investing in ecological infrastructure is empirically weak for three of the most invested-in South African catchments. (ii) There is a need for better baseline data collection, and monitoring during and after ecological infrastructure interventions to establish evidence for the benefits. (iii) We recommend that governments make funding available to support focused, applied and local empirical research to build an evidence base for the benefits of ecological infrastructure investments globally. (iv) Private sector investment could potentially complement and relieve constrained public sector funding; however, evidence of financial returns is required. (v) Finance to be able to implement ecological infrastructure interventions at scale could bring greater benefits to nature and people through more sustainable land-use.

Supporting data. The datasets supporting this article have been uploaded as part of the supplementary material.
Data accessibility. Data are made available in the electronic supplementary material.
Authors' contributions. A.J.R., P.B.H., K.E. and M.G.N. conceptualized the study. A.J.R. and P.B.H. carried out the analysis. A.J.R. drafted the manuscript, and P.B.H., M.G.N. and K.E. revised it. All authors gave final approval for publication and agree to be held accountable for the work performed therein.
Competing interests. We declare we have no competing interests.
Acknowledgements. We acknowledge the Danish Ministry of Foreign Affairs (DANIDA) for funding the research undertaken under the SEBEI (Socio-Economic Benefits of Investing in Ecological Infrastructure) Project, grant no: 17-M07-KU. We also thank all the stakeholders that participated in the research process.

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
