## [Peer Review File · Royal Society Open Science]

Review History

RSOS-201402.R0 (Original submission)

Review form: Reviewer 1

Is the manuscript scientifically sound in its present form?

No

Are the interpretations and conclusions justified by the results?

No

Is the language acceptable?

Yes

Do you have any ethical concerns with this paper?

No

Have you any concerns about statistical analyses in this paper?

No

Recommendation?

Major revision is needed (please make suggestions in comments)

Comments to the Author(s)

Review RSOS-201402

Title: Benefits of water-related ecological infrastructure investments to support sustainable land use: A review of evidence from critically water-stressed catchments in South Africa

OVERALL

The topic of this paper on empirical evidence for ecosystem service gains following ecological infrastructure interventions is an interesting one and topical. However, I have some key concerns about the conclusions drawn. First, it needs to be made clearer in the Abstract that this is a rapid evidence assessment of the literature in these three catchments using a new framework. I am not convinced that the evidence shows the headline results discussed, i.e. that ecological infrastructure interventions are largely ineffective. There are issues with the presentation of the results. Three catchments are chosen and I am unsure, but it seems that the results have been pooled over the three catchments? If so, this is a serious concern. It might be that the two southeastern catchments are sufficiently similar and that the results can be pooled, but we are told, but no description is forthcoming, that the eastern catchment is different. Therefore, it seems reasonable that different ecological infrastructure interventions might be more suitable for different ecosystems/climates/types of catchment, e.g. pristine, suburban, urban or lowland, upland. Showing the results by catchment would be superior. If the two south-eastern catchments are similar show one in the results and one in supplementary info.

More generally, from the results we see, it seems that a different, and I would argue perhaps more interesting in its nuance, is that specific ecological infrastructure interventions do produce the outcomes expected, including fire and revegetation either with irrigation or mature vegetation and more holistic restoration interventions. You can then also focus on the ecological infrastructure interventions that seem to fail – perhaps across catchments (info not provided) – suggesting there is something about these interventions that needs changing, e.g. the need for irrigation or monitoring, or that these single interventions do not work on their own.

ISSUES TO ADDRESS

1. ABSTRACT

It needs to be made clearer that this is a rapid evidence assessment of the empirical evidence around ecological infrastructure interventions in South African river ecosystems.

I am not sure the data does show no positive impact – see OVERALL comment above. Sure in some ecosystems there is no positive response but is this outcome more related to the fact that we would not expect positive change in these instances or that we would expect a negative change, e.g. removing IAPs with machinery and increased erosion? It seems to be that what this analysis shows is that two interventions might not be working as expected, namely Agricultural Retreat and Clearing IAPs and that the Combo project, i.e. combination restoration projects are best (also shown in Rose charts) and a handful of other interventions such as wetlands, reintroduction of fire, irrigation or planting mature vegetation also lead to positive outcomes. Indeed the importance of general restoration projects is highlighted as a special case that had to be created in the Lit Review.

2. CASE STUDY CATCHMENTS AND METHODS

I would first describe the catchments and in much more detail than that provided. Need to provide at a minimum the size of the catchment, inflows, key water users/uses, main ecosystems, type of catchment, i.e. pristine, urban, alpine, lowland. Info on Page 11, Lines 35-38 should be moved here, i.e. a description of invasives.

I would then have a separate Methods section outlining, in much more detail, the three steps: framework, rapid evidence assessment – trends in ecosystem service delivery and the type of

study (conceptual, empirical, modelled – I would put the type of study first or it could be placed in supplementary info with just summary stats in text, e.g. most studies were conceptual...), and casual loops.

It was unclear to me how the results were calculated, i.e. I was expecting to see them for each catchment but these are somehow generalised, how? Or are they pooled? At a minimum, we might expect to see the two southern catchments analysed together (but a case would have to be made to do this) and the eastern catchment analysed separately.

3. DISCUSSION

It might be the three discussion points you have discussed are the appropriate discussion points once you explain or address what might be issues with the results. I think the first paragraph at least needs to be clearer. Perhaps first discuss the types of papers, i.e. most conceptual and the need for more empirical or modelled papers and then what the results say. I believe the arguments are much more nuanced than you state in the paragraph.

More discussion of the causal loops part of the research is needed.

Limitations – ideally you would have a nearby control catchment without any intervention to ensure you are accounting for environmental change/climate change. Or you could expand this point in your Conclusion

4. CONCLUSIONS

These must be written in full paragraphs not bullet points.

5. TABLES 1 and 2

The tables did not fit on the page and therefore it was hard to comment on them fully.

6. MINOR (but necessary)

PAGE 3

Lines 30-37 can probably be deleted

Rename at Line 40 to simply Case Study Catchments

Line 46 to 47 – intriguing, what are the differences? See above under CASE STUDY

CATCHMENTS

Then add a sub-title Methodology here you can use some of the text from Lines 30-37 around the framework, rapid evidence assessment of the literature and use of causal loops.

Table 1 and 2 are not fully visible and therefore I

PAGE 4

Line 47 – irrigation seems to be an obvious other limiting factor and I see later on in Table 1, irrigation is mentioned so add it here, if it was part of your assessment.

PAGE 8

Lines 30 to 45 – it would be good to know the overlaps between your framework, the 12 ecological infrastructure interventions from the lit review, i.e. add a table.

Decision letter (RSOS-201402.R0)

Dear Dr Rebelo

The Editors assigned to your paper RSOS-201402 "Benefits of water-related ecological infrastructure investments to support sustainable land use" have now received comments from reviewers and would like you to revise the paper in accordance with the reviewer comments and any comments from the Editors. Please note this decision does not guarantee eventual acceptance.

Please submit your revised manuscript and required files (see below) no later than 21 days from today's (ie until the 19th of March). Note: the ScholarOne system will 'lock' if submission of the revision is attempted 21 or more days after the deadline. If you do not think you will be able to meet this deadline please contact the editorial office immediately.

on behalf of Professor Brian Reid (Associate Editor) and Agnieszka Latawiec (Subject Editor)
openscience@royalsociety.org

Associate Editor Comments to Author (Professor Brian Reid):
Comments to the Author:

Methodological issues are raised regarding pooling data from dissimilar catchments. This and the other comments raised need to be addressed.

Reviewer comments to Author:
Reviewer: 1

Comments to the Author(s)
Review RSOS-201402

Title: Benefits of water-related ecological infrastructure investments to support sustainable land use: A review of evidence from critically water-stressed catchments in South Africa

OVERALL

The topic of this paper on empirical evidence for ecosystem service gains following ecological infrastructure interventions is an interesting one and topical. However, I have some key concerns about the conclusions drawn. First, it needs to be made clearer in the Abstract that this is a rapid evidence assessment of the literature in these three catchments using a new framework. I am not convinced that the evidence shows the headline results discussed, i.e. that ecological

infrastructure interventions are largely ineffective. There are issues with the presentation of the results. Three catchments are chosen and I am unsure, but it seems that the results have been pooled over the three catchments? If so, this is a serious concern. It might be that the two southeastern catchments are sufficiently similar and that the results can be pooled, but we are told, but no description is forthcoming, that the eastern catchment is different. Therefore, it seems reasonable that different ecological infrastructure interventions might be more suitable for different ecosystems/climates/types of catchment, e.g. pristine, suburban, urban or lowland, upland. Showing the results by catchment would be superior. If the two south-eastern catchments are similar show one in the results and one in supplementary info.

More generally, from the results we see, it seems that a different, and I would argue perhaps more interesting in its nuance, is that specific ecological infrastructure interventions do produce the outcomes expected, including fire and revegetation either with irrigation or mature vegetation and more holistic restoration interventions. You can then also focus on the ecological infrastructure interventions that seem to fail – perhaps across catchments (info not provided) – suggesting there is something about these interventions that needs changing, e.g. the need for irrigation or monitoring, or that these single interventions do not work on their own.

ISSUES TO ADDRESS

1. ABSTRACT

It needs to be made clearer that this is a rapid evidence assessment of the empirical evidence around ecological infrastructure interventions in South African river ecosystems.

I am not sure the data does show no positive impact – see OVERALL comment above. Sure in some ecosystems there is no positive response but is this outcome more related to the fact that we would not expect positive change in these instances or that we would expect a negative change, e.g. removing IAPs with machinery and increased erosion? It seems to be that what this analysis shows is that two interventions might not be working as expected, namely Agricultural Retreat and Clearing IAPs and that the Combo project, i.e. combination restoration projects are best (also shown in Rose charts) and a handful of other interventions such as wetlands, reintroduction of fire, irrigation or planting mature vegetation also lead to positive outcomes. Indeed the importance of general restoration projects is highlighted as a special case that had to be created in the Lit Review.

2. CASE STUDY CATCHMENTS AND METHODS

I would first describe the catchments and in much more detail than that provided. Need to provide at a minimum the size of the catchment, inflows, key water users/uses, main ecosystems, type of catchment, i.e. pristine, urban, alpine, lowland. Info on Page 11, Lines 35-38 should be moved here, i.e. a description of invasives.

I would then have a separate Methods section outlining, in much more detail, the three steps: framework, rapid evidence assessment – trends in ecosystem service delivery and the type of study (conceptual, empirical, modelled – I would put the type of study first or it could be placed in supplementary info with just summary stats in text, e.g. most studies were conceptual...), and causal loops.

It was unclear to me how the results were calculated, i.e. I was expecting to see them for each catchment but these are somehow generalised, how? Or are they pooled? At a minimum, we might expect to see the two southern catchments analysed together (but a case would have to be made to do this) and the eastern catchment analysed separately.

3. DISCUSSION

It might be the three discussion points you have discussed are the appropriate discussion points once you explain or address what might be issues with the results. I think the first paragraph at least needs to be clearer. Perhaps first discuss the types of papers, i.e. most conceptual and the need for more empirical or modelled papers and then what the results say. I believe the arguments are much more nuanced than you state in the paragraph.

More discussion of the causal loops part of the research is needed.

Limitations – ideally you would have a nearby control catchment without any intervention to ensure you are accounting for environmental change/climate change. Or you could expand this point in your Conclusion

4. CONCLUSIONS

These must be written in full paragraphs not bullet points.

5. TABLES 1 and 2

The tables did not fit on the page and therefore it was hard to comment on them fully.

6. MINOR (but necessary)

PAGE 3

Lines 30-37 can probably be deleted

Rename at Line 40 to simply Case Study Catchments

Line 46 to 47 – intriguing, what are the differences? See above under CASE STUDY

CATCHMENTS

Then add a sub-title Methodology here you can use some of the text from Lines 30-37 around the framework, rapid evidence assessment of the literature and use of causal loops.

Table 1 and 2 are not fully visible and therefore I

PAGE 4

Line 47 – irrigation seems to be an obvious other limiting factor and I see later on in Table 1, irrigation is mentioned so add it here, if it was part of your assessment.

PAGE 8

Lines 30 to 45 – it would be good to know the overlaps between your framework, the 12 ecological infrastructure interventions from the lit review, i.e. add a table.

===PREPARING YOUR MANUSCRIPT===

===PREPARING YOUR REVISION IN SCHOLARONE===

Author's Response to Decision Letter for (RSOS-201402.R0)

See Appendix A.

Decision letter (RSOS-201402.R1)

Dear Dr Rebelo

On behalf of the Editors, we are pleased to inform you that your Manuscript RSOS-201402.R1 "Benefits of water-related ecological infrastructure investments to support sustainable land use" has been accepted for publication in Royal Society Open Science subject to minor revision in accordance with the referees' reports. Please find the referees' comments along with any feedback from the Editors below my signature.

Please submit your revised manuscript and required files (see below) no later than 7 days from today's (ie 06-Apr-2021) date. Note: the ScholarOne system will 'lock' if submission of the revision is attempted 7 or more days after the deadline. If you do not think you will be able to meet this deadline please contact the editorial office immediately.

on behalf of Professor Brian Reid (Associate Editor) and Agnieszka Latawiec (Subject Editor)
openscience@royalsociety.org

Associate Editor Comments to Author (Professor Brian Reid):

See the attached file for comments from Reviewer 2

===PREPARING YOUR MANUSCRIPT===

===PREPARING YOUR REVISION IN SCHOLARONE===

-- If you have uploaded ESM files, please ensure you follow the guidance at <https://royalsociety.org/journals/authors/author-guidelines/#supplementary-material> to include a suitable title and informative caption. An example of appropriate titling and captioning may be found at https://figshare.com/articles/Table_S2_from_Is_there_a_trade-off_between_peak_performance_and_performance_breadth_across_temperatures_for_aerobic_scops_in_teleost_fishes_/3843624.

Author's Response to Decision Letter for (RSOS-201402.R1)

See Appendix B.

Decision letter (RSOS-201402.R2)

Dear Dr Rebelo,

I am pleased to inform you that your manuscript entitled "Benefits of water-related ecological infrastructure investments to support sustainable land use" is now accepted for publication in Royal Society Open Science.

on behalf of Professor Brian Reid (Associate Editor) and Agnieszka Latawiec (Subject Editor)
openscience@royalsociety.org

Appendix A

Benefits of water-related ecological infrastructure investments to support sustainable land use: A review of evidence from critically water-stressed catchments in South Africa

Response to reviewers

OVERALL

The topic of this paper on empirical evidence for ecosystem service gains following ecological infrastructure interventions is an interesting one and topical. However, I have some key concerns about the conclusions drawn.

> Thank you for your input. The aim of this manuscript is to determine whether a locally relevant evidence base exists to support or substantiate the investments in ecological infrastructure that have been made in these three South African catchments. Our research question is: “is there evidence that ecological infrastructure interventions are delivering the proposed benefits?” Therefore, the focus here is on the state of the evidence base (both peer-reviewed and grey literature), and not measuring the efficacy of ecological infrastructure interventions themselves. We have adjusted the text throughout the abstract and introduction to make sure the aim is clearer so that readers do not make the same misinterpretation as the reviewer. Given our focus, this is not a rapid assessment, but a very thorough approach, assessing the state of the evidence base itself, including the development of a novel framework, a full literature review (considering not only the published literature, but also grey literature) and integrating both these into a systems dynamic model.

First, it needs to be made clearer in the Abstract that this is a rapid evidence assessment of the literature in these three catchments using a new framework.

> We agree that we could emphasize more strongly that we have developed a new framework for this type of work in the abstract, and have amended the abstract text to reflect this.

I am not convinced that the evidence shows the headline results discussed, i.e. that ecological infrastructure interventions are largely ineffective.

> Our key result is: “The evidence base for the benefits of investing in ecological infrastructure is limited for three of the most invested in South African catchments” (i.e. there has been little empirical research undertaken in the catchments despite the large investments made). We never state that “ecological infrastructure interventions are largely ineffective”; this is entirely out of the scope of this study. We have amended the abstract and introduction to make sure that there is no confusion that we focus on the state of the evidence base, and not the efficacy of these ecological infrastructure interventions.

There are issues with the presentation of the results. Three catchments are chosen and I am unsure, but it seems that the results have been pooled over the three catchments? If so, this is a serious concern. It might be that the two southeastern catchments are sufficiently similar and that the results can be pooled, but we are told, but no description is forthcoming, that the eastern catchment is different. Therefore, it seems reasonable that different ecological infrastructure interventions might

be more suitable for different ecosystems/climates/types of catchment, e.g. pristine, suburban, urban or lowland, upland. Showing the results by catchment would be superior. If the two south-eastern catchments are similar show one in the results and one in supplementary info.

> Due to our specific research question, the location of the interventions may be interesting, but these are not important in answering the question, which was phrased for strategic water source areas, not to investigate the effect of investments in different ecosystem types. We do agree that had our study been about the success of ecological infrastructure interventions, then considering ecosystem type, climate, soils, as well as socio-economic parameters would be critical.

More generally, from the results we see, it seems that a different, and I would argue perhaps more interesting in its nuance, is that specific ecological infrastructure interventions do produce the outcomes expected, including fire and revegetation either with irrigation or mature vegetation and more holistic restoration interventions.

> We trust that our amendments to the framing of the paper and our adjustments to the methodology will ensure that readers do not misinterpret the findings shown in our framework to be from the literature review conducted. We firstly hypothesised a framework and then mapped the literature onto these hypotheses. Also see last section of results.

You can then also focus on the ecological infrastructure interventions that seem to fail – perhaps across catchments (info not provided) – suggesting there is something about these interventions that needs changing, e.g. the need for irrigation or monitoring, or that these single interventions do not work on their own.

> Please see responses above.

ISSUES TO ADDRESS

1. ABSTRACT

It needs to be made clearer that this is a rapid evidence assessment of the empirical evidence around ecological infrastructure interventions in South African river ecosystems.

> Please see responses above. Given our aim of the manuscript (which we have now worked on making clearer in the abstract and introduction), this was not a rapid assessment of the evidence. We conducted an in-depth and detailed analysis of the evidence for these three catchments. We included both peer reviewed and grey literature.

I am not sure the data does show no positive impact – see OVERALL comment above.

> We never state that “the data shows no positive impact” in the manuscript. The reviewer is correct in that the data does show mostly positive impacts of EI interventions. For example we describe that most cases (96%) show positive impacts. However, for all interventions the evidence is mostly propositional and not empirically measured or modelled (244 of our 269 cases were conceptual). This is what we refer to as a limited or weak evidence base, which is our key finding of the manuscript. We

trust that our changes to the abstract and introduction assist the reviewer in understanding the results in this way.

Sure in some ecosystems there is no positive response but is this outcome more related to the fact that we would not expect positive change in these instances or that we would expect a negative change, e.g. removing IAPs with machinery and increased erosion? It seems to be that what this analysis shows is that two interventions might not be working as expected, namely **Agricultural Retreat** and **Clearing IAPs** and that the Combo project, i.e. combination restoration projects are best (also shown in Rose charts) and a handful of other interventions such as wetlands, reintroduction of fire, irrigation or planting mature vegetation also lead to positive outcomes. Indeed the importance of general restoration projects is highlighted as a special case that had to be created in the Lit Review.

> We cannot state that there is no positive response. We did not measure this. Please see responses above. However, the reviewer makes a good point that combining several ecological infrastructure interventions may be considered a more holistic approach, however we did not aim to explore this more, and this is out of the scope of this study. It would be interesting to look at this in the future: comparing single ecological infrastructure approaches to holistic approaches, and/or ecological restoration.

2. CASE STUDY CATCHMENTS AND METHODS

I would first describe the catchments and in much more detail than that provided. Need to provide at a minimum the size of the catchment, inflows, key water users/uses, main ecosystems, type of catchment, i.e. pristine, urban, alpine, lowland. Info on Page 11, Lines 35-38 should be moved here, i.e. a description of invasives.

> The catchments are indeed very interesting, both in terms of their similarities and their differences. We have written a full study site description in a report available on our website: <http://www.acdi.uct.ac.za/socio-economic-benefits-ecological-infrastructure-sebei>. However the critical point about these catchments for this particular study is the fact that they have been heavily invested in, in terms of ecological infrastructure interventions. This is why they were selected for this study. Further details are interesting but not critical to the research question, and would be an issue in terms of the word count imposed by the journal. However, that said, we have added a few extra sentences on ecology, climate, key water-users and key issues to this section.

In terms of the size of the catchment, and hydrology, please see Figure 1. These are hydrological “catchments” and therefore have no “inflows” (as shown in Fig 1).

We would prefer to not move the info on Page 11, Lines 35-38, as this is about the systems dynamic modelling, however have indicated instead that one of the many problems faced by this catchment is invasion by alien plants, and also particularly by alien trees.

I would then have a separate Methods section outlining, in much more detail, the three steps: framework, rapid evidence assessment – trends in ecosystem service delivery and the type of study (conceptual, empirical, modelled – I would put the type of study first or it could be placed in supplementary info with just summary stats in text, e.g. most studies were conceptual...), and casual loops.

> We have added a subtitle Methodology as the reviewer suggests in a comment below. We then include our descriptions of the three steps under this subtitle before going into the details per section. The first part of the methods is the catchment descriptions, followed by sections describing the framework, the literature review (the evidence assessment), and the systems dynamic model (causal loops).

It was unclear to me how the results were calculated, i.e. I was expecting to see them for each catchment but these are somehow generalised, how? Or are they pooled? At a minimum, we might expect to see the two southern catchments analysed together (but a case would have to be made to do this) and the eastern catchment analysed separately.

> Please see the detailed description of how this was done on page 5:19-33, and page 6: 3-24. Essentially, one database was composed, of which “catchment” was just one of the variables recorded. There is no need to analyse catchments separately as this was not part of our research question nor study design.

3. DISCUSSION

It might be the three discussion points you have discussed are the appropriate discussion points once you explain or address what might be issues with the results. I think the first paragraph at least needs to be clearer. Perhaps first discuss the types of papers, i.e. most conceptual and the need for more empirical or modelled papers and then what the results say.

> In the first paragraph of the discussion we follow the order of our results in summarising key findings. This is why we do not start with describing the fact that most studies were conceptual.

I believe the arguments are much more nuanced than you state in the paragraph.

> We are not sure to which paragraph the reviewer is referring and therefore it is difficult to make any revisions if needed.

More discussion of the causal loops part of the research is needed.

> While we include the causal loop approach in the paper as a way to bring our findings together it is not the main aim of the research and therefore given our word limit, we would prefer to keep the current focus of the discussion. We do however provide key findings on this specifically on page 16: 20-24, and it is discussed in quite a bit of detail on page 18: 7-38.

Limitations – ideally you would have a nearby control catchment without any intervention to ensure you are accounting for environmental change/climate change. Or you could expand this point in your Conclusion

> See responses above. We believe the reviewer has misunderstood the purpose of the study. The purpose is not to measure intervention success. This is not an experiment, and there is no need for a

control catchment. We hope our changes to the abstract and introduction have helped make sure the research purpose is now clear.

4. CONCLUSIONS

These must be written in full paragraphs not bullet points.

> The conclusions have been converted from summary points to a paragraph.

5. TABLES 1 and 2

The tables did not fit on the page and therefore it was hard to comment on them fully.

> We apologise for this, this was a formatting issue. Only the last column was cut-off, please check the latest upload for the full table.

6. MINOR (but necessary)

PAGE 3

Lines 30-37 can probably be deleted

> Thank you for your input. We have integrated this under a Methodology section as the reviewer suggests.

Rename at Line 40 to simply Case Study Catchments

> Renamed.

Line 46 to 47 – intriguing, what are the differences? See above under CASE STUDY CATCHMENTS

> The catchments are indeed very interesting, both in terms of their similarities and their differences. We have written a full study site description in a report available on our website: <http://www.acdi.uct.ac.za/socio-economic-benefits-ecological-infrastructure-sebei>. However the critical point about these catchments for this particular study is the fact that they have been heavily invested in, in terms of ecological infrastructure interventions. This is why they were selected for further study. Further details are interesting but not critical to the research question, and would be an issue in terms of the word count imposed by the journal.

Then add a sub-title Methodology here you can use some of the text from Lines 30-37 around the framework, rapid evidence assessment of the literature and use of causal loops.

> We have integrated the lines 30-37 under a new methodology section and described the phases that we used to do the research.

Table 1 and 2 are not fully visible and therefore I

> Our apologies, this is now fixed.

PAGE 4

Line 47 – irrigation seems to be an obvious other limiting factor and I see later on in Table 1, irrigation is mentioned so add it here, if it was part of your assessment.

> This paragraph details the assumptions we made when undertaking the assessment. These are factors we think should be made explicit. Irrigation was not assumed, but was explicitly categorised in the framework. Therefore it is not relevant to add it here.

PAGE 8

Lines 30 to 45 – it would be good to know the overlaps between your framework, the 12 ecological infrastructure interventions from the lit review, i.e. add a table.

> Please see the final column of Table 1, this is exactly what it details. Our apologies as this was the one column cut off in the proof.

Appendix B

RSOS-201402.R1

Review 2: Benefits of water-related ecological infrastructure investments to support sustainable land use: A review of evidence from critically water-stressed catchments in South Africa

Thanks for having the chance to read the revised paper. It is good to see the whole table and to read through the responses to the comments. I still have a couple of comments that if addressed would, I believe, further improve the paper.

-> We thank the reviewer for the helpful inputs to improve the paper.

I think part of my confusion with the paper's aims was that it is described in different ways, even after the review. In terms of the aim of the paper there seem to be at least 4 different ways of phrasing this in the text. I suggest you choose one and reiterate it. I personally prefer #3 or #4. I would edit #1 - What is the state of the evidence base for the benefits of investing in ecological infrastructure? i.e., is there evidence that the types of ecological infrastructure interventions in the study catchments can deliver the proposed benefits?

1. Page 8 of 42, Lines 38-42: "What is the state of the evidence base for the benefits of investing in ecological infrastructure? I.e., is there evidence for the study catchments that ecological infrastructure interventions are delivering the proposed benefits?"
2. Page 10 of 42, Lines 26-27 "We ask the question: is there evidence that ecological infrastructure interventions are delivering the proposed benefits?"
3. Page 12 of 42, Lines 24-26 "Therefore, we aimed to establish whether a rigorous evidence base of the benefits of ecological infrastructure interventions on ecosystem service provision existed to support the investments that have been made in these catchments."
4. Page 23 of 42, Lines 42-46 "We aimed to establish whether there is evidence available that ecological infrastructure interventions are delivering the proposed benefits in three well invested and researched South African catchments."

-> We have selected the wording for number 4 and changed the other three to match.

Keywords: I would evidence

-> We selected our keywords with SEO (Search Engine Optimisation) in mind and the word 'evidence' appears in the title.

Page 11 of 42, Lines 43-51 change to development of systems dynamic model – or change subtitle to Systems dynamic model from causal loop

We have edited this section to make the methodology more clear to show it is a conceptual systems dynamic model. This now links well to the text where we describe the methodology as using systems thinking and causal loop diagramming principles.

Page 15 of 42, Lines 10/11 – what type of relationship, inverse, positive?

-> Thank you, we have made this change. It is 'positive'.

Page 18 of 42, Lines 3-14 – this categorisation seems an important insight. If it is new, I would also mention it in the Abstract.

-> Thank you. Upon uploading the final version of the manuscript, we were notified of a 200-word limit for the abstract. It was 265 words in its present form. Therefore we have had to reduce the abstract in accordance with this limit. Thus we are unfortunately limited as to the level of detail we can add to our abstract.

Page 23 of 42 – For flow and to reduce repetition, make the second para (Lines 27-38) the first paragraph and integrate info from the first paragraph (Lines 5-25) in the 3 sub-headings, e.g. at Line 49 move the information from Lines 14-17

-> To improve flow, we have moved this summary of results to the end of the results section, and start the discussion with the paragraph suggested by the reviewer. The reviewer suggested that we integrate these results in the results section in the previous round of review comments.

Page 24 of 42, Line 19 – Split the long paragraph, i.e. start a new paragraph beginning “A review of nature-based solutions in ... then perhaps begin that paragraph with lines 23-26 “Although the ecological infrastructure investment evidence-base is weak limited for these three catchments, in most cases ecological infrastructure interventions were found to be beneficial.” Then contrast this to the UK paper by Dick et al (2019

-> We have made this change.